# Foam Synthesis of Nickel/Nickel (II) Hydroxide Nanoflakes Using Double Templates of Surfactant Liquid Crystal and Hydrogen Bubbles: A High-Performance Catalyst for Methanol Electrooxidation in Alkaline Solution

**DOI:** 10.3390/nano12050879

**Published:** 2022-03-07

**Authors:** Amani M. Bamuqaddam, Saba A. Aladeemy, Mohamed A. Ghanem, Abdullah M. Al-Mayouf, Nouf H. Alotaibi, Frank Marken

**Affiliations:** 1Chemistry Department, College of Science, King Saud University, Riyadh 11451, Saudi Arabia; 439204234@student.ksu.edu.sa (A.M.B.); sabaameen98@yahoo.com (S.A.A.); amayouf@ksu.edu.sa (A.M.A.-M.); nhalotaibi@ksu.edu.sa (N.H.A.); 2Chemistry Department, College of Applied Science, Taiz University, Taiz 6803, Yemen; 3K.A.CARE Energy Research and Innovation, Riyadh 11454, Saudi Arabia; 4Department of Chemistry, University of Bath, Bath BA2 7AY, UK; fm202@bath.ac.uk

**Keywords:** nickel/nickel hydroxide, nanoflakes, template, methanol oxidation

## Abstract

This work demonstrates the chemical synthesis of two-dimensional nanoflakes of mesoporous nickel/nickel (II) hydroxide (Ni/Ni(OH)_2_-NFs) using double templates of surfactant self-assembled thin-film and foam of hydrogen bubbles produced by sodium borohydride reducing agent. Physicochemical characterizations show the formation of amorphous mesoporous 2D nanoflakes with a Ni/Ni(OH)_2_ structure and a high specific surface area (165 m^2^/g). Electrochemical studies show that the electrocatalytic activity of Ni/Ni(OH)_2_ nanoflakes towards methanol oxidation in alkaline solution is significantly enhanced in comparison with that of parent *bare*-Ni(OH)_2_ deposited from surfactant-free solution. Cyclic voltammetry shows that the methanol oxidation mass activity of Ni/Ni(OH)_2_-NFs reaches 545 A/cm^2^ g_cat_ at 0.6 V vs. Ag/AgCl, which is more than five times higher than that of *bare*-Ni(OH)_2_. Moreover, Ni/Ni(OH)_2_-NFs reveal less charge transfer resistance (10.4 Ω), stable oxidation current density (625 A/cm^2^ g_cat_ at 0.7 V vs. Ag/AgCl), and resistance to the adsorption of reaction intermediates and products during three hours of constant-potential methanol oxidation electrolysis in alkaline solution. The high-performance electrocatalytic activity of Ni/Ni(OH)_2_ nanoflakes is mainly derived from efficient charge transfer due to the high specific surface area of the 2D mesoporous architecture of the nanoflakes, as well as the mass transport of methanol to Ni^2+^/Ni^3+^ active sites throughout the catalyst layer.

## 1. Introduction

Methanol is regarded as an attractive fuel to feed direct methanol fuel cells (DMFCs) because of the advantages of high energy density, reduced CO_2_ emission, low operating temperature, and ease of storage and transport [1,2,3]. Nevertheless, the DMFC suffers from several challenges that limit its performance in commercial markets due to the high price of platinum catalyst and the slow kinetics of the methanol oxidation reaction (MOR) as a result of catalyst surface poisoning [1,4,5,6,7]. Consequently, it is necessary to develop low-cost, non-noble, and efficient transition-metal electrocatalysts to improve electrocatalytic activity and durability for MOR [8,9,10].

Engineering the electrocatalyst architectures, particularly for nickel-based materials at the nanometer scale, shape, facets, and composition, significantly boosts catalytic activity for the oxidation of small organic molecules [4,11,12]. For instance, nickel-based nanostructured electrocatalysts such as Ni/Ni(OH)_2_ [13], Ni(OH)_2_-NiCoO_4_ [14], and NiO MOF/rGO [15] have gained substantial interest and revealed significant electrocatalytic performance with tolerance to CO poisoning during ethanol [16] and methanol [17,18] electrooxidations in an alkaline environment. Within this context, NiO crystalline nanoparticles, having uniform dispersion and deposited on pristine carbon nanotubes (CNTs) via atomic layer deposition, revealed significant activation from methanol oxidation due to enriched Ni(II)/Ni(III) redox-active sites [11].

Recently, heterostructures of Ni cores encapsulated within NiO shells and supported on carbon nanotubes (NiO-Ni/CNT) were shown to boost the electrocatalytic methanol oxidation reaction with an onset oxidation potential of 0.325 V vs. Hg/HgO and normalized mass activity of about 2.1 A/mg at 0.58 V vs. Hg/HgO, and the catalyst was stable for up to 300 cycles during methanol oxidation in alkaline solution [19]. Interestingly, a high-surface-area porous nickel hydroxide catalyst electrodeposited using a dynamic hydrogen bubble template was reported to exhibit durable catalytic methanol oxidation electroactivity in alkaline solution and achieved a current density of 181 mA/cm^2^ [20]. Wang et al. [21] demonstrated that a molecular catalyst of four-six-coordinated nickel hydroxide (NR-Ni(OH)_2_) nanoribbon resulted in a significantly lower onset potential of 0.55 V vs. RHE for the oxidation of methanol in alkaline solution. The enhanced methanol performance was related to the presence of charge-transfer orbitals for delocalized electrons near the Fermi level that efficiently enhance methanol oxidation to CO_2_ via a six-electron transfer reaction in alkaline media.

Very recently, Ni_3_S_2_ enriched with defects and supported on carbon nanofibers (CNFs) showed significant electroactivity and selectivity in the oxidation of methanol to valuable formate with the co-production of hydrogen at an applied potential of 1.37 to 1.62 V vs. RHE and current density of more than 700 mA/cm^2^ [12,22]. The Faradic efficiency of the process reached 99%, and the synergistic effect of Ni–OOH and SO_x_ active sites was responsible for selective oxidation of methanol to formate and cathodic hydrogen-intensive production. In related work, our research group reported nickel/nickel hydroxide nanoflake (Ni/Ni(OH)_2_-NF) catalysts obtained by chemical deposition and exfoliation through the self-assembly of a surfactant hexagonal template mixture. The nanoarchitecture of Ni/Ni(OH)_2_-NFs revealed a high mesoporous surface area of 135 m^2^/g and semi-crystalline Ni-incorporated α-Ni(OH)_2_ nanoflakes [23]. The obtained Ni/Ni(OH)_2_ nanostructures showed superior activity over the bare-Ni(OH)_2_ catalyst for electrooxidation of small organic molecules such as ethanol [23], urea [24], and glucose [25] in alkaline solution.

In this study, we developed a thin-film self-assembly of surfactant liquid crystal template (in a Petri dish) to prepare 2D mesoporous nanoflakes of Ni/Ni(OH)_2_ by chemical reduction and exfoliation of the confined nickel deposit by hydrogen foam generated by sodium borohydride reducing agent. The physicochemical properties of the obtained catalyst were evaluated by different techniques. The electrochemical performance, long-term stability, and charge transfer kinetics of the fabricated Ni/Ni(OH)_2_ nanoflake structure were examined for the electrocatalytic oxidation of methanol in alkaline conditions using electrochemical techniques, namely, cyclic voltammetry, current–time transients, and electrochemical impedance spectroscopy (EIS), respectively.

## 2. Materials and Methods

### 2.1. Chemicals and Materials

Nickel (II) nitrate hexahydrate (Ni(NO_3_)_2_.6H_2_O, 98.0%) and sodium borohydride (NaBH_4_, 97.0%) were acquired from LOBA Chemie (Mumbai, India). The nonionic surfactant polyethylene glycol octadecyl ether (Brij^®^78, C_18_H_37_(OCH_2_CH_2_)_20_OH, 97.0%), potassium hydroxide (KOH, 85.0%), and methanol (CH_3_OH, 99.8%) were obtained from Sigma-Aldrich. Carbon paper substrate (CP, SIGRACET^®^, grade GDL-24BC, SGL) was obtained from SGL Carbon (Chedde, France), and Nafion™ perfluorinated resin 5 wt.% solution of 45% water mixed in aliphatic alcohols was obtained from Merk. No further purification was carried out for the chemicals, and all solutions were prepared using deionized water (resistivity of 18.2 MΩ cm) obtained from a PURELAB^®^ Flex water distillation machine.

### 2.2. Nickel/Nickel Hydroxide Ni/Ni(OH)_2_ Nanoflake Synthesis

Figure 1 shows a schematic diagram for the synthesis of 2D nanoflakes of Ni/Ni(OH)_2_ material (Ni/Ni(OH)_2_-NFs) using the approach of thin-film self-assembly of Brij^®^78 surfactant mixed with nickel precursor solution. Firstly, the template mixture of nickel precursor was obtained by physically mixing Brij^®^78 surfactant (2.0 g, melted at 40 °C) with a solution of nickel nitrate (conc. 0.5 M, 10.0 mL) in a Petri dish (10 cm diameter). After that, the template mixture in the Petri dish was transferred to an ultrasonic bath for 20 min to obtain a homogeneous solution and then transferred to an oven at 25 ± 2 °C to evaporate the excess of water to reach a surfactant/water ratio of 40% to obtain the hexagonal template of surfactant liquid crystal. After that, an excess of sodium borohydride solution (NaBH_4_, 2.0 M, 10 mL) in a ratio of 1:4 (nickel nitrate/NaBH_4_) was sprayed over the nickel template mixture in the Petri dish.

Effervescence of hydrogen foam began to form upon the addition of NaBH_4_, as shown in Figure 1, and the template mixture gradually changed from green to black due to the commencement of the Ni/Ni(OH)_2_-NFs catalyst deposition. The Petri dish with black catalyst was maintained overnight to ensure the completion of the deposition reaction. Then, the surfactant was removed by washing the precipitated catalyst with hot distilled water (50 °C) five times under magnetic stirring. Lastly, the catalyst deposit was separated via centrifugation and placed in an oven overnight to dry at 50 °C. The control catalyst (*bare*-Ni(OH)_2_) was precipitated in the absence of the Brij^®^78 surfactant by reacting NaBH_4_ solution (2.0 M) with 1.0 M aqueous nickel nitrate solution.

### 2.3. Characterizations of Ni/Ni(OH)_2_ Catalysts

The nanostructure and morphology of the as-prepared Ni/Ni(OH)_2_ nanoflake catalyst were investigated by transmission electron microscope (TEM, JEM 2100F-JEOL, Tokyo, Japan) and scanning electron microscope (SEM, JSM-7600F; JEOL), respectively. The V-Sorb 2800 Porosimetry Analyzer was employed to measure the specific surface area of Ni/Ni(OH)_2_ catalyst, which was determined using the Brunauer–Emmer–Teller (BET) approach. In addition, the as-synthesized catalyst crystal profile was examined by MiniFlex-600 (Rigaku) X-ray diffraction (XRD) analysis with Cu K_α_ radiation (λ = 1.5418 A°) at operation conditions of 40 kV and 15 mA.

Cyclic voltammetry (CV), chronoamperometry (CA), and electrochemical impedance spectroscopy (EIS) characterizations were performed using AutoLab Potentiostatic Galvanostat PGSTAT302N (Metrohm). The electrochemical studies were executed using a glass cell with a three-electrode configuration with a thin sheet of platinum (0.5 cm × 0.5 cm) and Ag/AgCl as counter and reference electrodes, respectively. The working electrode was made of SIGRACET^®^ gas diffusion carbon paper (GDL-24BC, 1.0 cm × 1.0 cm) loaded with catalyst ink immersed in methanol and potassium hydroxide test solution. The Ni/Ni(OH)_2_ nanoflake ink was prepared by mixing 10 mg of the catalyst, 0.5 mL of deionized water, 0.5 mL of isopropanol, and 10 μL of Nafion in a glass vial and placed in an ultrasonic bath for 30 min. Catalyst ink with predefined loading was placed on one side of the GDL carbon paper using a micropipette and dried in an open atmosphere by hot air to obtain the working electrode. The obtained current was normalized to the catalyst loading and 1.0 cm^2^ electrode geometric area.

For comparison, electrochemical measurements of the reference (*bare*-Ni(OH)_2_) catalyst were also performed using the same procedure, and throughout this work, all test solutions were prepared using deionized water (DI; resistivity of 18.2 MΩ cm) obtained from a Milli-Q ultrapure water purification system.

## 3. Results

### 3.1. Catalyst Characterizations

As shown in Figure 1, the mesoporous nickel/nickel (II) hydroxide nanoflakes (Ni/Ni(OH)_2_-NFs) were synthesized using the modified approach with a surfactant thin-film template and sodium borohydride (NaBH_4_) reducing agent at room temperature (25 °C). During this synthesis procedure, nickel ions confined within the interstitial aqueous domain of the surfactant template were chemically precipitated by NaBH_4_ to instantaneously produce hydrogen gas, which exfoliated and fragmented the confined deposited catalyst layers in the form of nanoflakes. Figure 1 shows the morphology of the catalyst surface and fine nanostructure of the Ni/Ni(OH)_2_ nanoflakes, as characterized by scanning (SEM) and transmission (TEM) electron microscopes. The SEM images in Figure 1a reveal that the assembled Ni/Ni(OH)_2_ catalyst has a spherical microparticle morphology, and the microparticles themselves are formed from an array of overlapping nanometer-sized flakes. Moreover, the TEM micro-images in Figure 1b,c show that the 2D nanoflakes are very thin (thickness of about 10.0 nm) and strongly crumpled with nonuniform assembly and an open mesoporous channel. The elemental analysis, performed by energy-dispersive X-ray spectroscopy (EDX), as shown in Figure 1d, confirms that the nickel and oxygen contents (wt.%) are 47.33% and 52.67%, respectively, which correspond to a Ni/O mole ratio of 1:3 and are consistent with the hydrated Ni(OH)_2_ composition.

The Ni/Ni(OH)_2_-NF crystal structure was studied by X-ray diffraction (XRD). As revealed in Figure 2a, the Ni/Ni(OH)_2_-NFs exhibit broad and low-intensity diffraction peaks, which indicate the formation of an amorphous structure. According to JCPDS card no. 38-0715, the diffraction peaks observed at 2Θ = 12.07, 25.38, 33.68, 36.35, and 59.98° can be indexed to the (001), (002), (110), (111), and (300) diffraction planes, respectively, which are characteristic of the phase of the α-Ni(OH)_2_ crystal structure. The diffraction planes at 2θ = 44.55 and 70.98° can be assigned to the (111) and (220) planes of the pure nickel structure. These physicochemical characterization results confirm that the catalyst has mesoporous nanoflake morphology and Ni/α-Ni(OH)_2_ composition and structure [26].

The N_2_ adsorption–desorption isotherm curves of Ni/Ni(OH)_2_-NF catalyst and corresponding *bare*-Ni(OH)_2_ are shown in Figure 2b. According to the IUPAC classification of adsorption–desorption isotherms [27], the isotherms of Ni/Ni(OH)_2_-NFs and *bare*-Ni(OH)_2_ both show a type IV (H_3_) shape with a N_2_ gas capillary condensation hysteresis loop in a P/P° range of 0.45–1.0.

The N_2_ adsorption–desorption isotherm confirms that the Ni/Ni(OH)_2_-NFs architecture possesses a mesoporous structure with disordered slit and lamellar shapes, which is consistent with the Ni/Ni(OH)_2_-NFs nanoflake morphology [28]. The BET-specific surface area of Ni/Ni(OH)_2_-NFs obtained from the N_2_ adsorption–desorption isotherms was estimated to be 165 ± 5.0 m^2^/g, which is significantly higher (about seven times) than that of the *bare*-Ni(OH)_2_ counterpart (23.5 m^2^/g). The results indicate the presence of a typical mesoporous material throughout the nanoflake architecture, which is consistent with that mentioned in the literature for mesoporous nickel hydroxide architecture [24,28].

Figure 2c shows the FT-IR spectra of the obtained Ni/Ni(OH)_2_-NFs alongside the *bare*-Ni(OH)_2_ catalyst. The FT-IR spectra of both Ni/Ni(OH)_2_-NFs and *bare*-Ni(OH)_2_ catalysts show two characteristic peaks around 3472 and 1621 cm^−1^, which are assigned to OH^−^ stretching and H-O-H bending vibration modes of the adsorbed water molecules, respectively. However, the OH^−^ stretching peak of Ni/Ni(OH)_2_-NFs at 3472 cm^−1^ is strongly visible and more defined, suggesting the presence of more surface active sites (surface area) available for the adsorption of hydroxyl ions and water molecules. In addition, the bands at around 654 and 465 cm^−1^ correspond to the O-Ni-O bond and Ni-O-H bending vibrations, respectively, confirming the formation of the Ni(OH)_2_ surface. The bands around 2932, 1621, 1456, 1268, and 1079 cm^−1^ can be attributed to the vibration modes of intercalated nitrate ions and the residual adsorbed hydrocarbon species at the catalyst surface.

### 3.2. Electrochemical Characterization of Ni/Ni(OH)_2_-NF Catalyst

Figure 3a shows the electrochemical behavior, as obtained by successive multicyclic voltammograms (CVs), of the 100 µg/cm^2^ Ni/Ni(OH)_2_-NF electrode alongside *bare*-Ni(OH)_2_ and blank GDL carbon electrodes in a KOH solution (2.0 M) at a 50 mV s^−1^ potential scan rate.

During the first cycle, the anodic/cathodic redox peaks observed at potentials of 0.465/0.180 V vs. Ag/AgCl for Ni/Ni(OH)_2_-NFs and 0.480/0.165 V for *bare*-Ni(OH)_2_ can be assigned to the Ni(II)/Ni(III) redox couple as a result of the reaction (Ni(OH)_2_ + OH^−^ → NiOOH + H_2_O + e^−^) [29,30]. As cycling continues, very small changes in redox peak potential and current are observed, and the redox peak separation (ΔE = E_anodic_ − E_cathodic_) reaches 0.320 and 0.285 V for *bare*-Ni(OH)_2_ and Ni/Ni(OH)_2_-NFs, respectively, indicating that the Ni(II)/Ni(III) redox reaction is more stable and more reversible with the Ni/Ni(OH)_2_-NF catalyst. As shown above, the X-ray diffraction result suggests the Ni/Ni(OH)_2_-NF catalyst has a non-stoichiometric Ni and α-Ni(OH)_2_ structure, and because of the reaction with air/moisture, an intercalated buffer layer (1–2) of NiO can form (Ni/NiO/α-Ni(OH)_2_), which is electrochemically converted to nickel oxyhydroxide (NiOOH) according to the reported mechanism by Medway et al. [31].

The Ni(II)/Ni(III) redox peak current density is significantly affected by the Ni/Ni(OH)_2_-NF loading, as illustrated in Figure 3b. The cyclic voltammograms displayed in Figure 3b show the effect of various catalyst loadings (10, 50, and 100 μg) alongside a comparison with *bare*-Ni(OH)_2_ (100 μg, blue line) in a 2.0 M KOH solution at a scan rate of 50 mV s^−1^. It can be seen that at similar catalyst loadings, the Ni(II)/Ni(III) redox peak current of the Ni(OH)_2_-NF electrode is significantly higher than that of *bare*-Ni(OH)_2_, and the current density of the redox peak steadily increases when the Ni(OH)_2_-NF loading increases. The results indicate that the electrochemical active surface area (ECSA) of Ni(OH)_2_-NFs is superior due to the unique nanoflake architecture with enhanced porosity, which increases the rate of the redox process and facilitates the mass transport of OH− ions to the catalyst surface [32].

The electroactive surface area (ECSA) of Ni(OH)_2_-NFs can be estimated using the equation ECSA = Q/(mq), where Q is the charge under the reduction peak of the Ni(OH)_2_/NiOOH redox reaction, q is the charge required for Ni(OH)_2_ monolayer formation (equivalent to 257 μC/cm), and m is the catalyst mass, [33]. Herein, the ECSA of Ni/Ni(OH)_2_-NFs and *bare*-Ni(OH)_2_ catalyst is estimated at 354.3 and 23.5 m^2^/g, respectively. The ECSA of Ni/Ni(OH)_2_-NFs is 15 times higher than that of *bare*-Ni(OH)_2_, which is a significant difference and attributed to the presence of nanoflakes with a highly mesoporous ultrathin architecture, in agreement with the BET results above.

Figure 3c shows the cyclic voltammograms of 100 µg/cm^2^ Ni(OH)_2_ nanoflake catalyst and illustrates the effect of increasing KOH electrolyte concentrations on the electrochemical performance of the catalyst. A significant shift in the onset potential of the oxidation peak is observed as the KOH concentration increases from 0.1 to 2.0 M, which is attributed to the Nernst effect, in which the potential is inversely proportional to the hydroxyl ion concentration.

The nature of the redox process at the Ni(OH)_2_-NF electrode surface was investigated by observing the effect of increasing the scan rate from 5 mV s^−1^ to 50 mV s^−1^, as shown in Figure 3d. As soon as the scan rate increases, a slightly positive shift in the anodic peak potential is observed, while the cathodic peak moves in the negative direction. In addition, a linear relation is obtained between the redox peak currents and the square root of the scan rate (inset in Figure 3d), which is evidence that the formation of the β-NiOOH reaction is controlled by ion diffusion; this result agrees well with previously reported results [23,25].

To evaluate the electrocatalytic performance towards the methanol oxidation reaction (MOR) in an alkaline medium, the CVs of Ni(OH)_2_-NF and bare-Ni(OH)_2_ catalysts were recorded at 50 mV s^−1^ in 2.0 M KOH solution containing 0.5 M methanol, and the results are shown in Figure 4a. As illustrated in Figure 4a, the mesoporous Ni(OH)_2_-NF catalyst shows a substantial anodic current enhancement reaching 54.5 mA/cm^2^ at 0.6 V vs. Ag/AgCl, which is five times higher than that observed for the *bare*-Ni(OH)_2_ catalyst, confirming the superior electrocatalytic activity of the Ni(OH)_2_-NF catalyst towards methanol oxidation in alkaline solution. Additionally, the mesoporous Ni(OH)_2_-NFs display a methanol oxidation onset potential of 0.330 V vs. Ag/AgCl, which is consistent with the potential of Ni^2+^/Ni^3+^ oxidation of the Ni(OH)_2_-NF catalyst in the absence of methanol. Figure 4c shows the CVs of different loadings of Ni(OH)_2_-NFs in the presence of 0.5 M methanol and 2.0 M KOH.

The results reveal that methanol oxidation currents at the mesoporous Ni(OH)_2_-NF electrode gradually increase with increasing catalyst loading due to the enhancement of Ni^2+^/Ni^3+^ active sites, which are required for methanol adsorption and oxidation in 2.0 M KOH. To identify the reaction mechanism of methanol oxidation at the Ni(OH)_2_-NF catalyst, CVs were performed at various scan rates from 5.0 to 50 mVs^−1^ in 2.0 M KOH containing 0.5 M methanol, and the results are shown in Figure 4d. Clearly, the CVs show that the methanol oxidation current increases with the scan rate. As shown in Figure 4d (inset), a linear plot between peak current density (i_peak_) and the square root of the scan rate is obtained, and the line intercepts with the current axis at about 30.5 mA/cm^2^ (R^2^ = 0.9892). This indicates that a mixed kinetic–diffusion-controlled methanol oxidation process occurs at the Ni(OH)_2_-NFs catalyst in alkaline conditions.

The mechanism of methanol oxidation at the Ni(OH)_2_-NFs electrode can be adapted according to the EC-type mechanism described by Fleischman et al. [33], in which the redox couple of Ni^2+^/Ni^3+^ species is generated during the anodic electrooxidation of Ni(OH)_2_ in alkaline solution according to Equation (1).
Ni(OH)_2_ + OH^−^ → NiOOH + H_2_O + e^−^(1)

Then, methanol chemically reacts with NiOOH active sites via Equation (2) and regenerates Ni(OH)_2_.
n NiOOH + CH_3_OH → n Ni(OH)_2_ + Products(2)

The observed decrease in the cathodic peak current, which is around 0.25 V vs. Ag/AgCl in the presence of methanol (Figure 4a), is associated with the consumption of NiOOH species in the chemical reaction with methanol and the regeneration of Ni(OH)_2_ according to Equation (2). Interestingly, this cathodic peak current decreases as the methanol concentration increases (Figure 4b) due to the depletion of Ni^2+^/Ni^3+^ active sites, while its current increases when the catalyst loading increases (Figure 4c). This confirms that the methanol oxidation reaction follows the EC-type mechanism [34].

Nyquist plots were generated to investigate the electrode/electrolyte interface charge resistance of electrochemical methanol oxidation. Figure 5a shows EIS Nyquist plots and the equivalent circuit recorded in 2.0 M KOH containing 0.5 M methanol at 100 µg/cm^2^ Ni(OH)_2_-NFs, as compared to *bare*-Ni(OH)_2_ electrodes, at an applied potential of 0.6 V vs. Ag/AgCl and frequency of 100 kHz to 0.1 Hz. Figure 5b illustrates the Nyquist plots of the Ni(OH)_2_-NF catalyst at different applied potentials from 0.35 to 0.60 V vs. Ag/AgCl. The Nyquist profile displays a very small and depressed arc in the high-frequency zone, while a large semicircle in the low-frequency zone can be observed at all studied potentials. As shown in Figure 5, the small semicircle in the high-frequency region can be attributed to the charge transfer characteristic of methanol oxidation at the electroactive Ni(OH)_2_/NiOOH sites, while the large semicircle in the low-frequency domain is associated with the impedance of the intermediates’ adsorption process.

The Nyquist plots in Figure 5a of the *bare*-Ni(OH)_2_ and Ni/Ni(OH)_2_-NF electrodes can be adequately modeled with the equivalent circuit shown in Figure 5a (inset). The fitted equivalent circuit parameters are defined as the electrolyte resistance (R_s_), the charge transfer resistance (R_1_), and the capacitance of the double layer at the catalyst/electrolyte interface (C_1_), while R_2_ is related to the rate of intermediate adsorption resistance, and Q_1_ is the equivalent adsorption capacitance of methanol oxidation at the Ni(OH)_2_/NiOOH interface. Table 1 clearly shows that R_s_, representing the electrolyte resistance, remains almost constant with an average value of 7.50 Ω. At an applied potential of 0.50 V or higher, the methanol charge transfer resistance R_1_ is less than 0.480 Ω for the mesoporous Ni/Ni(OH)_2_ nanoflakes, which is less than that of the *bare*-Ni(OH)_2_ electrode (1.46 Ω), indicating that superior methanol oxidation kinetics take place at the Ni/Ni(OH)_2_-NFs electrode. However, the charge transfer R_1_ significantly increases as the oxidation potential changes from 0.50 to 0.35 V vs. Ag/AgCl. This confirms that the kinetics of methanol electrooxidation at the Ni/Ni(OH)_2_ nanoflakes are enhanced at a higher potential, while, at a lower potential, the oxidation of the Ni/Ni(OH)_2_ to NiOOH species is predominant, as shown by the cyclic voltammetry results.

Table 1 reveals that, for the Ni/Ni(OH)_2_-NF electrode, the R_2_ values of intermediate adsorption resistance are significantly lower (1.24 Ω) in comparison to the *bare*-Ni(OH)_2_ electrode (26.8 Ω) and decrease as the applied potential becomes more positive. This decrease in R_2_ is related to the higher tolerance of mesoporous Ni/Ni(OH)_2_-NFs to the adsorption of methanol oxidation intermediates due to the availability of Ni^2+^/Ni^3+^ active sites.

The long-term stability of a catalyst is one of the important requirements of methanol fuel cell application. Figure 6a shows long-term oxidation for 100 multicycles using 100 µg/cm^2^ Ni/Ni(OH)_2_-NF at 50 mVs^−1^ in 0.5 M methanol in 2.0 M KOH. As shown in Figure 6a, after 100 cycles over a potential range of −0.200 to 0.700 V vs. Ag/AgCl, the methanol oxidation current density remains relatively stable, indicating that the mesoporous Ni/Ni(OH)_2_-NF catalyst is extremely stable during the methanol oxidation reaction. Figure 6b shows the chronoamperometric (current/time) transient during the long-term oxidation of 0.5 M methanol in 2.0 M KOH at different applied potentials (0.400, 0.500, and 0.700 V vs. Ag/AgCl) using different loadings of Ni/Ni(OH)_2_-NFs in comparison to the *bare*-Ni(OH)_2_ catalyst. The chronoamperometric measurements confirm that the methanol oxidation current density obtained at the Ni/Ni(OH)_2_ nanoflake electrode remains very stable and significantly increases upon increasing either the applied potential or the catalyst loading over an extended oxidation process of 200 min. Moreover, the current density at 0.700 V vs. Ag/AgCl shows that the methanol oxidation mass activity of the Ni/Ni(OH)_2_-NF catalyst (current density/catalyst mass) reaches about 625 A/cm^2^ g_cat_, which is more than seven times higher than the parent *bare*-Ni(OH)_2_ catalyst, which is consistent with the cyclic voltammetry results.

The outstanding stability and enhanced methanol oxidation activity exhibited by the Ni/Ni(OH)_2_-NF catalyst can be attributed to the presence of a mesoporous network of overlapping Ni/Ni(OH)_2_ nanoflakes with high surface area and Ni^2+^/Ni^3+^ active sites required for methanol adsorption and oxidation [13,17,23,25]. In addition, the methanol oxidation current stability indicates the tolerance and resistance of Ni/Ni(OH)_2_-NFs to passivation by the adsorption of the intermediates and products of the methanol oxidation reaction [7,35].

The electrochemical activity of our mesoporous Ni/Ni(OH)_2_ nanoflake catalyst was evaluated and compared with previously reported catalysts. Table 2 displays a comparison of characteristic methanol oxidation parameters, namely, onset potential, stability, and mass activity (current/cm^2^ g_cat_), between our mesoporous Ni/Ni(OH)_2_ nanoflake catalyst and cutting-edge nickel-based electrocatalysts reported in the literature. For instance, our Ni/Ni(OH)_2_ nanoflake catalyst showed significantly higher methanol oxidation mass activity (545 A/cm^2^ g_cat_ at 0.6 V vs. Ag/Cl) and stability than most of the cutting-edge nickel-based catalysts, such as NiCo_2_O_4_/SS (50 A/cm^2^ g_cat_) [35], NiO-NS@NW/NF (∼55 A/cm^2^ g_cat_) [36], NiCo_2_O_4_-rGO (∼63 A/cm^2^ g_cat_) [37], Ni_3_S_2_-CNFs/CC (∼120 A/cm^2^ g_cat_) [22], and a nanoribbon structure (NR-Ni(OH)_2_) (180 A/cm^2^ g_cat_) [21]. In addition, with the advantage of a simple synthesis approach in a one-pot template mixture and under room conditions, the Ni/Ni(OH)_2_-NF catalyst exhibits a comparable methanol oxidation mass activity to that of the nickel metal–organic framework supported on reduced graphene oxide (NiO-MOF/rGO, 540 A/cm^2^ g_cat_) [34].

## 4. Conclusions

This work investigated a novel foam synthesis approach using Ni/Ni(OH)_2_ nanoflakes prepared via double templates of surfactant liquid crystal thin film and hydrogen bubbles induced by sodium borohydride reducing agent. The physicochemical characterizations revealed the formation of well-defined amorphous 2D Ni/Ni(OH)_2_ nanoflakes with a specific surface area of 165 m^2^/g. The electrochemical study shows that the mesoporous Ni/Ni(OH)_2_ nanoflakes possess significant activity for methanol oxidation in alkaline solution with a mass activity of 625 A/cm^2^ g at 0.7 V vs. Ag/AgCl, which is seven times higher than that observed at the bare-Ni(OH)_2_ catalyst. Moreover, the mesoporous Ni/Ni(OH)_2_ nanoflakes exhibited much lower charge transfer resistance, and multicycling and current–time transient measurements revealed superior durability, with no change in methanol oxidation current density after an extended methanol oxidation process of 100 cycles or 200 min. The outstanding methanol oxidation activity and stability and enhanced methanol oxidation activity obtained by the Ni/Ni(OH)_2_-NF catalyst can be ascribed to the formation of the 2D hierarchical mesoporous nanoflake architecture of overlapping Ni/Ni(OH)_2_ nanoflakes with high surface area and the Ni^2+^/Ni^3+^ active sites required for methanol adsorption and oxidation. In addition, the methanol oxidation current stability indicates that Ni/Ni(OH)_2_-NFs are tolerant and resistant to passivation by the adsorption of methanol oxidation intermediates and products.

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
