# Peer review of "Foam Synthesis of Nickel/Nickel (II) Hydroxide Nanoflakes Using Double Templates of Surfactant Liquid Crystal and Hydrogen Bubbles: A High-Performance Catalyst for Methanol Electrooxidation in Alkaline Solution"

_nanomaterials, 2022, doi:10.3390/nano12050879_

Round 1

Reviewer 1 Report

Ni/Ni(OH)2-NFs of high specific surface area and improved catalytic performance for methanol electrooxidation are revealed. The experiment data support the conclusions on the whole.

Seveal points need to explain.

1 Line 11, Yamen, where?

2 Line 94, how much NaBH4(2.0 M) was used? What is the ratio of Ni/Ni(OH)2 in Ni/Ni(OH)2-NFs. How to obtain Ni/Ni(OH)2-NFs in varied ratio of Ni/Ni(OH)2 and how about the effect. 

3  Figure 2(a), the peaks for Ni(OH)2 phase were not marked.
    Figure 2(b), what is Ni/Ni(OH)2-bulk. 

4  Figure 5(a) seems problem.

5 Expression

   Line 46, 51, 65, the space before the sentences.
   Line 55,  molecular catalyst, redundant.
   Line 88, Scheme 1shows.
   Line 266, 276, 280,292,297, there is no space for the first sentence. 

Author Response

The authors are grateful to the reviewer for spending the time to review the manuscript and giving comments which improve the work quality.

1- Line 11, Yamen, where?

Sorry for mistyping, corrected to Yemen

2- Line 94, how much NaBH4(2.0 M) was used? What is the ratio of Ni/Ni(OH)2 in Ni/Ni(OH)2-NFs. How to obtain Ni/Ni(OH)2-NFs in varied ratio of Ni/Ni(OH)2 and how about the effect.

Many thanks for this valuable comment. We performed many experiments of optimization using different concentrations of sodium borohydride and found that the concentration ratio of 1:4 of nickel precursor to sodium borohydride produced the most efficient catalyst with the highest surface area. It would be difficult to control the ratio of metallic nickel and nickel hydroxide (Ni/Ni(OH)2) of the produced nanoflakes catalyst because the metallic nickel surface spontaneously oxidized to form Ni(OH)2 at ambient conditions. The XRD analysis in Fig. 2a shows evidence of the presence of non-stoichiometric of metallic Ni and Ni(OH)2 deposit, However, this will be fully oxidized to Ni(OH)2 upon immersion in the KOH solution during performing the electrochemical measurements. The discussion on page 5 clarified this point.  

3 - Figure 2(a), the peaks for Ni(OH)2 phase were not marked.

The diffraction peaks of the Ni(OH)2 phase are labeled.

    Figure 2(b), what is Ni/Ni(OH)2-bulk.

Sorry for the mistyping, we mean the catalyst deposited in absence of surfactant (bare-Ni/Ni(OH)2). The legend in Fig. 2b is corrected.

4 -Figure 5(a) seems a problem.

In Fig. 5 we corrected the Figure legend mislabeling and the text font.

5- Expression

Line 46, 51, 65, the space before the sentences.

The space was removed from the mentioned sentences.

   Line 55,  molecular catalyst, redundant.

The word was removed.

   Line 88, Scheme 1shows.

The space added.

   Line 266, 276, 280,292,297, there is no space for the first sentence.

Sorry for the mistyping errors, all the above-mentioned expressions are corrected. 

Reviewer 2 Report

I recommend using spectral methods such as XPS, Raman and FT-IR spectroscopy to study catalysts.

Author Response

I recommend using spectral methods such as XPS, Raman, and FT-IR spectroscopy to study catalysts.

Many thanks for the suggestion. The FT-IR catalysts characterization is included in Fig. 2c and discussed explained on pages 5 and 6.  However, the accessibility to XPS and Raman techniques is very limited at the moment and we plan to perform such characterizations in the future.